# Direct-Acting Antivirals for HCV Treatment in Decompensated Liver Cirrhosis Patients: A Systematic Review and Meta-Analysis

**DOI:** 10.3390/jpm12091517

**Published:** 2022-09-15

**Authors:** JiHyun An, Dong Ah Park, Min Jung Ko, Sang Bong Ahn, Jeong-Ju Yoo, Dae Won Jun, Sun Young Yim

**Affiliations:** 1Gastroenterology and Hepatology, Hanyang University College of Medicine, Guri 04763, Korea; 2Division of Healthcare Technology Assessment Research, National Evidence-Based Healthcare Collaborating Agency (NECA), Seoul 04933, Korea; 3Department of Internal Medicine, Nowon Eulji Medical Center, Eulji University College of Medicine, Seoul 01830, Korea; 4Department of Internal Medicine, Soonchunhyang University Bucheon Hospital, Bucheon 14584, Korea; 5Department of Internal Medicine, Hanyang University College of Medicine, Seoul 04763, Korea; 6Department of Internal Medicine, Korea University College of Medicine, Seoul 02841, Korea

**Keywords:** chronic hepatitis C, decompensated liver cirrhosis, direct-acting antiviral, protease inhibitor

## Abstract

DAA therapy is known to clear hepatitis C virus infection in patients with decompensated cirrhosis (DC). However, the safety and benefits of DAA in DC remain unclear, especially with the use of protease inhibitors (PI). Therefore, we evaluated the efficacy and clinical safety of DAA in DC patients and observed whether there was a discrepancy between PI-based and non-PI-based treatment. We searched Ovid-Medline, Ovid-EMBASE, Cochrane Library, and three local medical databases through October 2021 to identify relevant studies on the clinical safety and effectiveness of DAA in DC patients. The outcomes were sustained virologic response (SVR), overall mortality, the incidence rate of hepatocellular carcinoma (HCC), adverse events, improvement or deterioration of liver function, and delisting from liver transplantation (LT). Two independent reviewers extracted the data from each study using a standardized form. The pooled event rate in DC patients and relative effect (odds ratio (OR)) of PI-treated versus non-PI-based DAA in DC patients were calculated using a random-effects model. In patients with DC, the SVR rate was 86% (95% CI 83–88%), the development of HCC 7% (95% CI 5–9%), and mortality 6% (95% CI 4–8%). Improvement in liver function was observed in 51% (95% CI 44–58%) of patients, and 16% (95% CI 5–40%) were delisted from LT. PI-based treatment showed a similar rate of serious adverse events (23% vs. 18%), HCC occurrence (5% vs. 7%), and mortality (5% vs. 6%) to that of non-PI-based DAA treatment in DC patients. HCC occurrence and mortality rates were low in patients with DC following DAA treatment. PI-based treatment in DC patients was relatively safe when compared to non-PI-based treatment. Overall, DAA improved liver function, which may have allowed for delisting from LT.

## 1. Introduction

It is estimated that 71 million people worldwide live with hepatitis C virus (HCV) [1], and 0.6% of the Korean population is chronically infected with HCV [2]. HCV infection is an important issue since 70% of untreated patients develop chronic HCV infection, and the risk of cirrhosis ranges from 15 to 30% within 20 years [1]. However, the progression to cirrhosis is often clinically silent, and some patients are unaware of the HCV infection. Asymptomatic liver cirrhosis may ultimately lead to decompensated cirrhosis (DC) with an annual risk of 3–5% and hepatocellular carcinoma (HCC). DC is a life-threatening condition with a mortality rate of 70% within 5 years, and liver transplantation is the only treatment available to avoid death [3]. 

In the interferon (IFN) era, treatment was limited to patients with either chronic hepatitis C (CHC) or compensated cirrhosis owing to safety concerns. IFN-based treatment in DC is associated with lower sustained virologic response (SVR) and increased risk of sepsis and hepatic encephalopathy, leaving patients with DC without tolerable treatment options [4,5]. The introduction of direct-acting antivirals (DAAs) has revolutionized the treatment of chronic HCV infection, and the efficacy and safety of DAA have been proven even in patients with compensated cirrhosis [6,7,8,9,10].

However, according to the U.S. Food and Drug Administration (FDA) report, some patients with compensated cirrhosis treated with protease inhibitors (PI) based DAA, such as paritaprevir/ritonavir/ombitasvir, developed jaundice and rapidly progressed to liver failure within 1–4 weeks of initiation of treatment [11]. A multicenter cohort study in Israel found that seven patients treated with paritaprevir/ritonavir/ombitasvir developed decompensation within 1–8 weeks of initiation of treatment, and one of these patients died [12]. Therefore, paritaprevir/ritonavir/ombitasvir ± dasabuvir is contraindicated in all patients with compensated cirrhosis because of concerns about hepatotoxicity. Additionally, data on the use of simeprevir in patients with Child–Turcotte–Pugh (CTP) class B cirrhosis are limited. In a phase 2 study comprising 40 patients (19 CTP class A and 21 CTP class B patients) treated with simeprevir, sofosbuvir, and daclatasvir for 12 weeks, the mean pharmacokinetic exposure to simeprevir was 2.2-fold greater after 8 weeks of treatment in CTP B than A [13,14]. It was higher in CTP class B than in CTP class A cirrhosis patients. Although all patients achieved SVR12, grade 3 or 4 bilirubin elevations were observed in 18% and 5% of the patients, respectively, although they were not associated with an increase in alanine transaminase (ALT) or the need for drug discontinuation.

Very recently, Torgerson et al. reported that PI-based DAAs are safe and do not increase the risk of hepatic decompensation compared to non-PI DAAs [15]. However, the number of decompensated patients included in that study was unclear. Patients with mild or well-compensated cirrhosis were included. Further studies are needed to confirm the safety and efficacy of PI-based DAAs.

Studies evaluating the efficacy of DAA in DC have been conducted. However, controversy exists regarding the treatment of patients with DC. In this study, we aimed to evaluate the efficacy and clinical safety of DAA in DC patients and to analyze whether PI-based DAA in DC patients resulted in a worse prognosis than non-PI-based treatment based on the results that have been published.

## 2. Materials and Methods

The review followed the Preferred Reporting Items for Systematic Reviews and Meta-analysis (PRISMA) guidelines [16].

### 2.1. Data Sources and Searches

We conducted literature searches of CHC treated with DAA in Ovid-MEDLINE, Ovid-Embase, Cochrane Register of Controlled Trials (CENTRAL), and three local Korean databases for articles published until 31 October 2021. As the first DAAs were approved by the FDA in 2011 [17], studies published before 2010 were excluded. The included DAAs were sofosbuvir, sofosbuvir and ribavirin, simeprevir, daclatasvir; ledipasvir, glecaprevir, pibrentasvir, elbasvir, grazoprevir, dasabuvir, velpatasvir, and ombitasvir/paritaprevir/ritonavir. The search terms were index terms and text words related to CHC or DAA. A search filter was applied and recommended by the Scottish Intercollegiate Guidelines Network to efficiently identify randomized controlled trials and observational studies. We applied no language limitations in the electronic database search, which was restricted to studies involving humans. Detailed search strategies for each database are provided in the Supplementary Material. The bibliographies of relevant articles were searched to identify additional publications. The protocol for this review was registered in advance in the International Prospective Register of Systematic Reviews (CRD42021241245). The review followed the PRISMA guidelines and the Meta-Analysis of Observational Studies in Epidemiology (MOOSE) checklist.

### 2.2. Definition

DC was defined by the presence of Child–Pugh B or C cirrhosis, ascites, hepatic encephalopathy, hepatic hydrothorax, and variceal hemorrhage, while those without complications as mentioned above and with Child–Pugh A cirrhosis were grouped as compensated cirrhosis (CC) patients.

SVR was defined as undetectable HCV RNA in the blood 12 weeks after antiviral therapy completion. Adverse events (AEs) were defined as any new symptoms during the treatment period independent of the requirement for dose reduction or treatment discontinuation. Serious AEs were adopted for analysis when they were mentioned in the study. Re-compensation of decompensated liver cirrhosis was defined as the restoration of cirrhosis status to Child–Pugh A, decrease in Model for End-Stage Liver Disease (MELD) score, or portal hypertension.

The use of PI in more than 20% of DC patients was classified as the PI-based treatment group, while those treated with less than 20% were categorized as the non-PI-based treatment group.

### 2.3. Study Selection

Studies reporting DAA patients with chronic HCV cirrhosis were considered eligible for inclusion. No restrictions on the subjects’ age, biological sex, or viral genotype were included in the study. All DAA regimens used worldwide were considered in the meta-analysis and were not limited to those approved by specific governments or institutions. Various prospective and retrospective studies, including randomized controlled trials, nonrandomized clinical trials, case-control studies, case-series studies, and cohort studies, have been included. Two reviewers (SYY and JHA) independently reviewed titles and abstracts. Full-text documents were independently examined after screening the titles and abstracts.

### 2.4. Data Extraction

The two reviewers who conducted the study selection independently extracted data from the selected studies into a standardized form, including (1) study characteristics: authors, year of publication, study location, design, and setting; (2) study population: number of compensated and decompensated liver cirrhosis patients, age, HCV genotype, and prior IFN treatment; (3) intervention: DAAs; and (4) outcome: SVR, improvement in liver function, AEs, discontinuation of DAA, development of HCC, death, and delisting from liver transplantation. Any disagreement between the two reviewers was resolved through discussion with a third reviewer (DAP or DWJ).

### 2.5. Risk of Bias Assessment

The risk of bias was assessed in selected studies using the Risk of Bias for Nonrandomized Studies (RoBANS) ver 2.0. Further information on the process and results of the risk of bias assessment are provided in the Appendix A. Two or three demonstrations of risk assessment were conducted. The risk of bias assessment was independently performed by two reviewers (SSY, JHA), and any discrepancy was resolved through a discussion with a third reviewer (JJY or SBA).

### 2.6. Statistical Analysis

The primary outcomes of this study were pooled event rate of (1) SVR; (2) serious adverse events (SAEs); (3) discontinuation of DAA; (4) HCC occurrence; (5) improvement in liver function; (6) delisting from liver transplantation; and (7) death in patients with DC. For dichotomous outcomes, the odds ratios (ORs) were calculated and reported with 95% confidence intervals (CIs) between PI-based DAA and non-PI-based DAA in DC patients. The pooled event rate was calculated in studies with only patients with DC. Heterogeneity among studies was initially determined by individual forest plots and later confirmed by Cochran’s Q statistic (*p* < 0.10, I^2^ ≥ 50%). Given the variability of the patients’ characteristics within the studies, the random-effects model was always applied as a conservative approach to all variables, regardless of I^2^ statistical data. Sensitivity and subgroup analyses were used to investigate the sources of heterogeneity and the factors that affected the magnitude of the effect. We prespecified and conducted subgroup analyses according to the specific characteristics of the study methods, study populations, and interventions. Egger’s test and funnel plot were used to detect publication biases associated with the variables used in ≥10 studies. Statistical analyses were performed with Review Manager, version 5.3 (RevMan, Copenhagen: The Nordic Cochrane Center) and the “meta” and “metafor” packages on the R version 3.6.3 (R Foundation for statistical software, version 3.6.3).

## 3. Results

### 3.1. Study Characteristics

Based on the database searches, 13,185 records were identified using a systematic review (Figure 1). After removing duplicates, 2803 records were checked for titles or abstracts, and 1690 studies were excluded after a thorough examination, of which 60 studies met the inclusion criteria after full-text review (Figure 1). No relevant articles were identified from the reference lists of review articles or meta-analyses.

Sixty studies including patients with HCV cirrhosis were conducted in 14 different countries, and nine (22%) studies were based on Asian data. Thirty studies were prospective cohort studies, and thirty were retrospective cohort studies. Forty studies included a history of prior treatment with IFN, and fifteen studies included PI-based treatment. Of the 60 studies, 41 included both DC and CC patients, while nine included DC patients only. Details of the study, including patient characteristics, types of DAA, and observed outcomes, are described in Table 1.

### 3.2. Risk of Bias Graph

The risk of bias is shown in Appendix A. Except for the selection of participants, most categories showed a low risk of bias, whereas comparability of patients and selective outcome reporting showed some proportion of unclear risk of bias.

### 3.3. Efficacy and Safety of DAA in DC Patients

Forty-nine studies, including 7886 patients with DC, included data on SVR following DAA treatment. Pooled analysis showed that the SVR rate was 86% (95% CI: 0.83–0.88) (Figure 2A). Patients were divided according to age, previous history of IFN treatment, use of PI-based DAAs, HCV genotypes, study design, industrial sponsorship, and study region, and for all of them, the SVR did not differ (Table 2).

We also analyzed the safety of DAAs in patients with DCs. The pooled rate for AEs and SAEs were 55% and 22%, respectively (n = 13 studies, 95% CI: 0.31–0.77 and n = 12 studies, 95% CI: 0.13–0.36, respectively) (Table 3, Figure 2B,C). AEs that resulted in drug discontinuation were further analyzed, and the pooled rate reached 6% (n = 11 studies, 95% CI: 0.04–0.08). The pooled rate for the development of HCC was 7% (n = 14, 95% CI: 0.05–0.09), and the overall mortality was 6% (n = 28, 95% CI: 0.04–0.08) in DC patients (Figure 2D,E).

When the efficacy of DAA in DC was compared to that of CC patients based on 30 studies, patients with decompensated cirrhosis showed a lower SVR rate (OR 0.43, 95% CI: 0.34–0.54) than CC patients. Furthermore, the protective effect of DAA on HCC development and mortality was also less significant in DC patients than in CC patients (n = 6, OR 2.67, 95% CI: 1.88–3.79 and n = 11, OR 6.14, 95% CI: 4.24–8.89, respectively) (Appendix AA–C).

### 3.4. Improvement in Liver Functions and Delisting from Liver Transplantation

Since improvement in liver function is an important endpoint in patients with DC, a thorough analysis was performed in patients with decompensation only to observe the effect of DAA on improvement or deterioration of liver function. Most studies (n = 17) defined changes in liver function using MELD, while eleven used changes in the Child–Pugh class, four used both, and eight observed clinical signs such as portal hypertension. Evaluation of improvement in liver function was conducted based on 34 studies that showed a pooled event rate of 51% (95% CI: 0.44–0.58) (Table 3). Further analysis was performed using six studies to observe whether the efficacy of DAA led to delisting from liver transplantation (LT) following re-compensation from DC. The positive impact of DAA allowed 16% (n = 6, 95% CI: 0.05–0.4) of the studied patients to be removed from the LT lists. However, deterioration of liver function could not be avoided in 16% (95% CI: 0.12–0.21) of DC patients when the analysis was performed according to 20 studies with heterogeneity of I^2^ > 70% for both outcomes (Table 3, Figure 3A–C).

In addition, the efficacy of DAA in DC was compared with that in CC patients, where DC patients showed a significantly higher rate of improvement in liver function (n = 8, OR 3.2, 95% CI: 1.29–7.95) than that of CC patients (Appendix AA). Moreover, the deterioration of liver function was comparable between the DC and CC patients (n = 3, OR 0.8, 95% CI: 0.47–1.38) (Appendix A).

### 3.5. Efficacy and Safety of PI-based DAA in DC Patients

Since the efficacy of PI-based DAA has not been fully studied in DC patients, its safety was compared to that of non-PI-based DAAs. The PIs included in this study were glecaprevir, grazoprevir, paritaprevir, and simeprevir.

The pooled SVR for 6315 PI-based DAA-treated DC patients was 85% (95% CI: 0.75–0.91), which did not differ from the non-PI-based treated group (n = 7866), with an SVR of 86% (95% CI: 0.84–0.89) (Table 2). In addition, PI did not have a significant impact on either AEs or SAEs, with pooled rates of 49% vs. 58% and 23% vs. 18% in PI- and non-PI-based treatments, respectively (Table 4). Patients treated with a PI-containing regimen (n = 2244) were more likely to be associated with the deterioration of liver function with a pooled event rate of 22% (95% CI: 0.20–0.25), while non-PI-based regimens (n = 2661) showed a pooled event rate of 14% (95% CI: 0.09–0.2). In contrast, the improvement in liver function was comparable between these two groups: 49% vs. 51% (n = 204, 95% CI: 0–1 and n = 3496, 95% CI: 0.42– 0.59) with a similar pooled mortality rate of 5% vs. 6% (n = 690, 95% CI: 0.04–0.07 and n = 5094, 95% CI: 0.04–0.09) (Table 4).

## 4. Discussion

The clinical impact of DAAs on DC was thoroughly assessed in the current study. The pooled SVR rate of DC (86%) is in accordance with previous major studies reporting an SVR rate of 83% in patients who received sofosbuvir–velpatasvir, a pan-genotypic DAA, 87% in patients receiving ledipasvir and sofosbuvir plus ribavirin, and 78–86% in those who received any DAA [5,9,21,24]. In comparison to that of CC, DC patients showed a lower SVR rate with an OR of 0.43. Although the pooled event rate of SVR was not as high as that of CC patients (>90%), DAA can be recommended for DC patients, as reaching SVR may lead to a better prognosis in DC patients.

The present study is the first to evaluate the pooled SVR rate and clinical events occurring in DC patients following DAA treatment. Unlike the SVR rate in the IFN era, which differed between Asian and Western countries owing to different proportions of HCV genotypes infected by patients that resulted in variable response rates [73], this was not observed in our studies. SVR was not associated with age or HCV genotype, while the presence of liver decompensation seems to be the only factor that affects the SVR rate. Based on many clinical trials, the effect of DAAs has been validated in both CHC and cirrhosis patients for all genotypes [74,75,76].

The occurrence of HCC and mortality are important outcomes in DC patients after HCV eradication following DAA treatment. We analyzed this important outcome and found the pooled mortality rate related to DAA and the development of HCC to be 6% and 7%, respectively, in DC patients. Furthermore, drug discontinuation following AEs was relatively low (6%) and did not lead to high mortality, as observed in the above results.

A study by Kumada et al. showed that the cumulative incidence rate of liver-related mortality confined to DC was lower in the DAA group (39.6%) than in the non-DAA group (50.6%), supporting the benefit of DAA in DC patients [33]. The large difference in survival rate between our meta-analysis and the study by Kumada et al. can be explained by different cut-off values for follow-up duration, where most of the follow-up duration in our studies was between 6 and 12 months following SVR; patients in the study by Kumada et al. were persistently followed up until death or loss. Another study by Cheung et al. showed a mortality rate of 9.9% over 15 months in DAA-treated patients, and 5.4% developed liver cancer [26]. In accordance with these results, overall DAA-related mortality and development of HCC were low in DC, which indicates that long-term clinical benefit following viral clearance can be expected. However, the effect of DAA did not lead to the same prognosis as that of CC. Since most of the studies included in our analysis had a follow-up duration of less than one year after SVR, the growth of HCCs could have been radiologically undetectable at treatment baseline rather than de novo development. Therefore, the presence of advanced liver disease at treatment initiation is the main factor predicting long-term outcomes, regardless of SVR or Aes of DAAs [24]. Therefore, early diagnosis of HCV infection is warranted before disease progression.

Another important issue in DC is the aggravation of liver function, which affects both survival and quality of life. First, we addressed the efficacy and safety of PI-based DAA in decompensated CHC patients. Pis are metabolized by the liver, and patients with impaired liver function are subsequently exposed to elevated serum PI concentrations. Although there are limited studies including patients with decompensated liver cirrhosis in clinical trials, the real-world efficacy, and safety of PI have been reported [15,21,49]. The improvement in liver function was comparable between those treated with PI and those who did not, with a similar rate of Aes and mortality between these two groups, demonstrating that PI-based regimens can be considered for DC patients. This is an important finding because the PI-containing regimen voxilaprevir/velpatasvir/sofosbuvir is approved for pangenotypic treatment after DAA failure, which is currently indicated for CC patients only where our study may provide evidence for considering treatment in DC patients when primary treatment fails.

When the improvement in liver function was assessed in overall DC patients, we observed an improvement in 51% of these patients. Effective antiviral treatment with suppression of ongoing hepatic inflammation may inhibit long-term cirrhosis-associated complications, and achievement of SVR may lead to improvement in hepatic function, as reflected by CTP and MELD scores. The improvement in liver function was more prominent in DC patients than in CC patients, most probably owing to the improvement in ascites, hepatic encephalopathy, and liver function represented by bilirubin and international normalized ratio. This could have led to the delisting of liver transplantation in 16% of DC patients treated with DAA, which could eventually lead to a decrease in mortality. In accordance with these findings, the number of liver transplantations in patients with HCV-related DC has been reported to have rapidly decreased in the DAA era compared to that in the IFN era [58,59,64].

Our study had several limitations. First, the number of studies on PI-based DAA in DC patients was small, with inconsistent proportions of PI-based DAAs due to the limited number of trials that included DC patients; therefore, analysis for every single PI regimen was not available. Second, the heterogeneity was moderate in the SVR rate and improvement in liver function. Several factors may have contributed to such variance, such as the study design, selective process of data, and degree of liver disease progression, which may have resulted in straying from the intended study design. Another limitation of this study is the lack of defined follow-up duration for analysis of mortality and HCC occurrence, rendering varying duration of follow-up after SVR, which cannot be adjusted by applying a cut-off for follow-up duration due to the small number of evidence in subgroup analysis. We should acknowledge that most of the evidence in this systematic review is based on observational studies that are subject to potential biases.

## 5. Conclusions

Despite these challenges and limitations, DAA is highly effective and well tolerated in decompensated liver cirrhosis patients, traditionally a hard-to-treat population. The PI-based DAAs appeared to be relatively safe without increasing the mortality rate compared to that of non-PI-based treated patients. However, comparative analysis with compensated LC demonstrated that the clearance of HCV in decompensated LC patients did not prevent death or HCC occurrence at a similar rate as compensated liver cirrhosis, as the underlying liver disease, liver function, and patient comorbidities are important factors. Nevertheless, the improvement in liver function was definite and more prominent in decompensated LC patients than in compensated LC patients, indicating that decompensated patients have a greater need for treatment, which eventually leads to improved overall survival. We believe that this systematic review may present the risks and benefits of DAA in decompensated patients based on the accumulated results reported by many countries where this is an area of interest in hepatology.

## Figures and Tables

**Figure 1 jpm-12-01517-f001:**
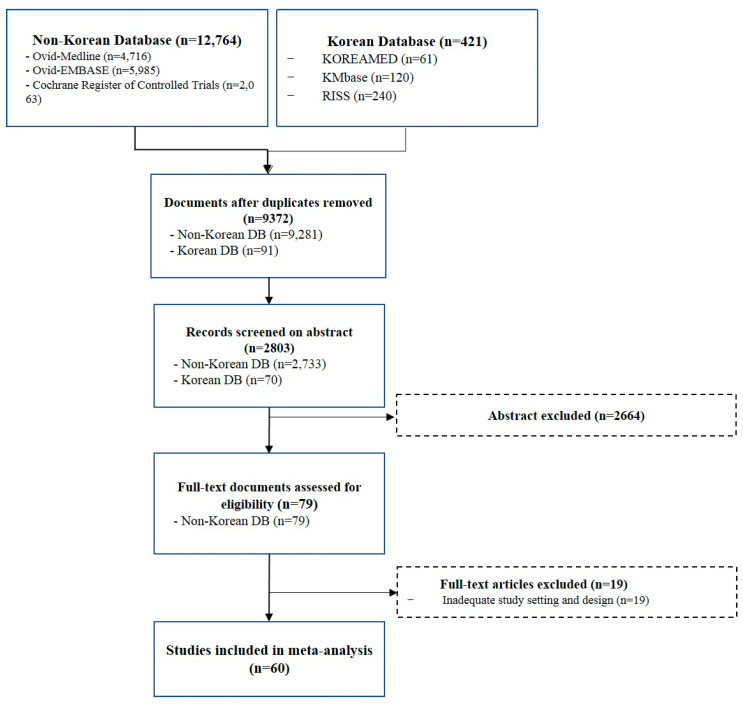
Flowchart of study selection for the systematic review and meta-analysis.

**Figure 2 jpm-12-01517-f002:**
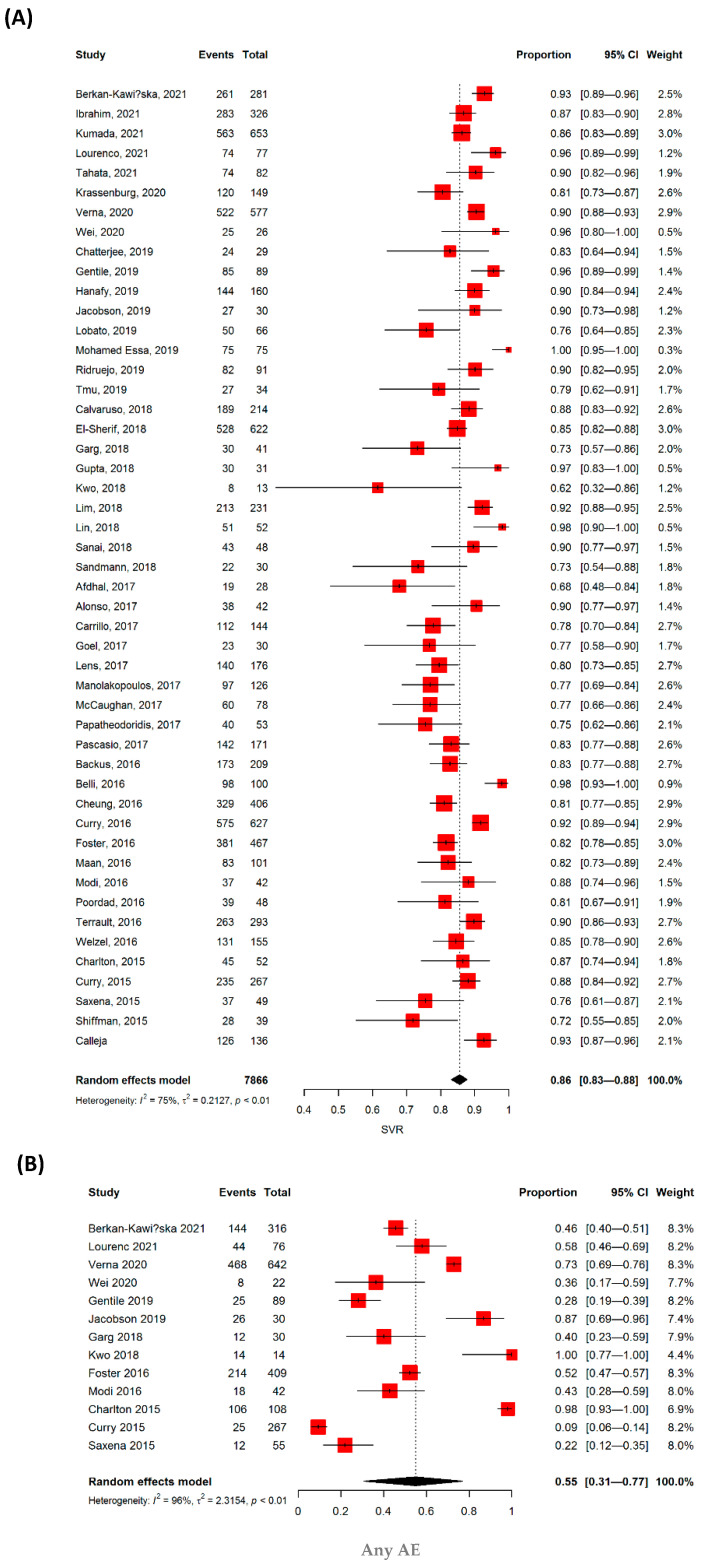
Estimated event rate of (**A**) the sustained virologic response, (**B**) any adverse events, (**C**) serious adverse events, (**D**) development of hepatocellular carcinoma, and (**E**) mortality in decompensated liver cirrhosis patients treated with direct-acting antiviral (DAA).

**Figure 3 jpm-12-01517-f003:**
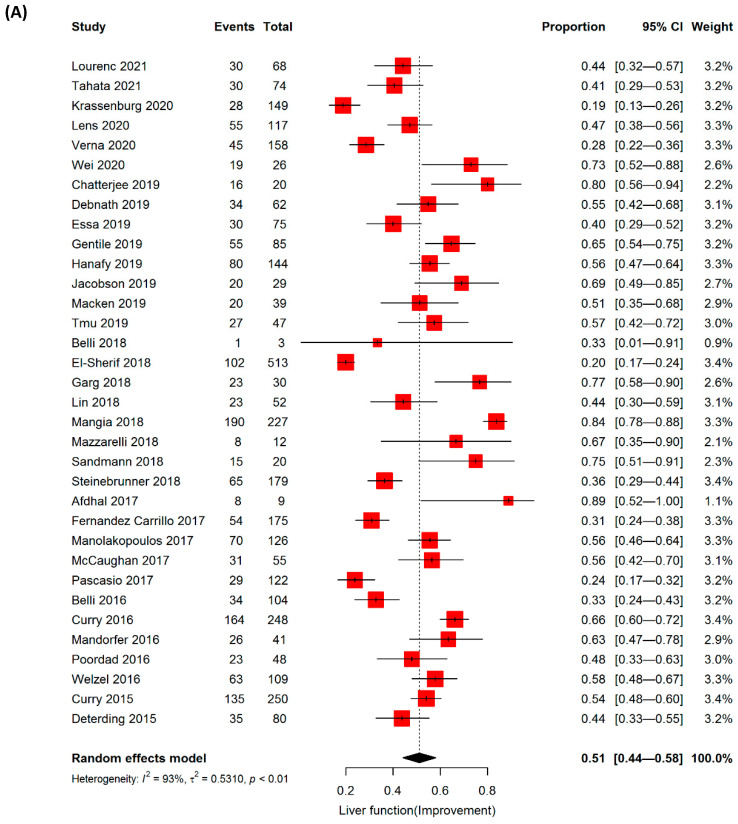
Estimated event rate of (**A**) improvement in liver function, (**B**) delisting from liver transplantation, and (**C**) worsening in liver function of decompensated liver cirrhosis patients treated with direct-acting antiviral (DAA).

**Table 1 jpm-12-01517-t001:** General characteristics of included studies (n = 60).

Author, Year	Study Location	Publication Type	Study Design	Total Patients	Age (Years)	Gender (Male)	Prior Treatment (%)	CC	DC	GT	DAA	PI-Based (≥20%)	FU Period	References *
Afdhal, 2017	Worldwide	Original	Prospective	50	55 (43–69)	76%	68%	18	32	1–4	SOF/RBV	No	NA	[18]
Alonso, 2017	Spain	Original	Retrospective	208	55 ± 8	82%	47%	166	42	3	SOF/LDV ± RBV, SOF/DCV ± RBV	No	12 wks	[19]
Backus, 2016	China	Original	Retrospective	6542	61.4 ± 6.2 (24.5–90.8)	96%	24%	6333	209	1	LDV/SOF ± RBV, PTV/r/DSV/RBV	No	NA	[20]
Berkan-Kawińska, 2021	Europe	Original	Prospective	2713	59 (50–65)	54%	41%	2397	316	1–6	LDV/SOF ± RBV, OBV/PTV/r ± DSV ± RBV, GRZ/EBR ± RBV, GLE/PIB, SOF/VEL ± RBV, SOF/DCV/RBV, SOF/IFN/RBV, SOF/RBV, ASV/DCV	Yes	At least 12 wks	[21]
Calleja, 2017	Spain	Original	Retrospective	3325	59.5 (21–87)	55%	45%	2362	136	1	OMV/PTV/r + DSV ± RBV, SOF/LDV ± RBV	No	24 or 36 wks	[22]
Calvaruso, 2018	Italy	Original	Prospective	2249	65.4 ± 10.7	57%	56%	2035	214	1–4	SOF/RBV, SOF/LDV ± RBV, SOF/DCV ± RBV, SOF/SIM ± RBV, OBV/PTV/r ± RBV, OBV/PTV/r/DSV ± RBV	No	14 (6–24) mo	[23]
Carrillo, 2017	Spain	Original	Retrospective	739	55 (36–80)	67%	60%	564	175	1/3	SOF/SIM, SOF/DCV, SOF/LDV OBV/PTV/r/DSV, SIM/DCV	Yes	12 wks	[24]
Chatterjee, 2019	India	Abstract	Prospective	50	55	62%	28%	21	29	1/3	SOF/LDV/DCV, SOF/VEL ± RBV	No	2 yrs	[25]
Cheung, 2016	UK	Original	Prospective	406	54 (28–79)	NA	NA	70	406	1/3	SOF/LDV ± RBV, SOF/DCV ± RBV	No	15 mo	[26]
Deterding, 2015	Germany	Original	Prospective	80	57 ± 9 (range 38–79)	59%	68%	45	34	1–4	SOF/RBV 56, SOF/SIM ± RBV, SOF/DCV ± RBV	NA	12 wks	[27]
Garg, 2018	India	Original	Prospective	63	47.2 ± 11.51 (32–75)	56%	0	33	30	3	SOF/LDV ± RBV	No	15 mo	[28]
Goel, 2017	India	Original	Prospective	160	45 (18–75)	39%	10%	31 (CHC79)	51	3	SOF/RBV, SOF/DCV, SOF/DCV/RBV	No	12/24 wks	[29]
Gupta, 2018	India	Original	Prospective	490	38.9 ± 12.7	57%	12%	120 (CHC339)	31	1–4	SOF/RBV, SOF/RBV/IFN, SOF/DCV, SOF/DCV/RBV	No	12 wks	[30]
Ibrahim, 2021	Egypt	Original	Retrospective	601	50.54 ± 12.82	37%	18%	275	326	4	DAA	NA	12 wks	[31]
Jacobson, 2019	USA	Original	Prospective	40	58.3 ± 7	57%	37%	10	30	1	EBR/GZR	Yes	12 wks	[32]
Krassenburg, 2020	Worldwide	Original	Retrospective	868	59 (54–65)	64%	12%	719	149	1–6	NS3/4 NS5A NS3/4 NS5B NS3/4 NS5A/B NS5B NS5A/B	No	28 (IQR 20–36)	[8]
Kumada, 2021	UK and Japan	Original	Prospective	364	54 (48–59)	72%	0	50	314	1–4	LDV/SOF ± RBV, SOF/DCV ± RBV	No	1.75 (0.71–3.05) yrs	[33]
Kwo, 2018	USA	Original	Retrospective	77	61 (34–79)	78%	56%	63	14	1–3	DCV/SOF ± RBV	No	24 wks	[34]
Lens, 2017	Spain	Original	Retrospective	922	72 (65–90)	58%	48%	746	176	1–4	SOF/RBV, LDV/SOF ± RBV, SOF/SIM ± RBV, DCV ± RBV, DCV/SIM ± RBV, PTV/OBV/RBV, OBV/PTV/DSV ± RBV	NA	12 wks	[35]
Lens, 2020	Spain	Original	Prospective	226	60 (53–69)	53%	NA	179	47	1–5	LDV/SOF ± RBV, SOF/SIM ± RBV SOF/DCV ± RBV	NA	24, 96 wks	[36]
Lim, 2018	USA	Original	Prospective	634	>65 (153 (24%)	66%	100%	383	251	1	LDV/SOF ± RBV	No	12 wks	[37]
Lobato, 2019	Brazil	Original	Prospective	3939	58 ± 10	60%	NA	3703	236	1–6	SOF/DCV, SOF/SIM, SOF/LDV, OBV/PTV/r/DSV ± RBV, SOF/RBV/PEG-IFN	NA	12, 24 mo	[38]
Maan, 2016	Worldwide	Original	Retrospective	433	57.8 ± 8.7	64%	65%	319	114	1–5	PI, DAA/RBV	Yes	12 wks	[39]
Macken, 2019	UK	Original	Prospective	1448	54 (47–60)	73%	41%	1344	104	1/3	OBV/PTV/r/DSV ± RBV SOF/LDV ± RBV, SOF/DCV ± RBV, SOF/PEG/RBV	NA	12 wks	[40]
Mandorfer, 2016	Austria	Original	Retrospective	120	52.6 ± 1.2	73%	NA	60	60	1–4	SOF/RBV, SOF/SIM, SOF/DCV, SOF/LDV, SIM/DCV	NA	12 wks	[41]
Mangia, 2018	Worldwide	Abstract	Prospective	1545	59 (26–86)	68%	NA	1318	227	NA	DAA	NA	53 (<1–144) wks	[42]
Mazzarelli, 2018	UK	Original	Retrospective	113	>65	53%	61%	101	12	1–4	SOF/LDV ± RBV, SOF/DCV ± RBV, SOF/RBV, OBV/PTV/r ± DSV ± RBV	NA	38 wks (12–132)	[43]
Papatheodoridis, 2017	Greece	Abstract	Retrospective	604	57 ± 11	58%	67%	386 (CHC158)	60	1–5	SOF/SIM ± RBV, SOF/DCV ± RBV, SOF/LDV ± RBV, 3D ± RBV, SOF/RBV ± pegIFNa, SOF/DCV ± RBV, 2D/RBV	Unknown	NA	[44]
Pascasio, 2017	Spain	Original	Retrospective	171	54 (51–61)	81%	49%	17	154	1,3,4	SOF/RBV, SOF/DCV ± RBV, SOF/SIM ± RBV, SOF/LDV ± RBV, 2D 3D ± RBV	No	NA	[45]
Poordad, 2016	USA	Original	Prospective	60	58 (19–75)	63%	60%	12	48	1–6	DCV/SOF/RBV	No	24 wks	[46]
Ridruejo, 2019	Argentina	Original	Prospective	906	60 ± 12	52%	55%	486	91	1–4	SOF/DCV ± RBV	No	22.3 mo	[47]
Sanai, 2018	Arab	Original	Prospective	213	59.6 ± 12.1	41%	40%	165	48	4	SOF/LDV ± RBV	No	12 wks	[48]
Saxena, 2015	USA	Original	Retrospective	156	61 (58–64)	61%	55%	101	55	1	SOF/SIM ± RBV	Yes	12 wks	[49]
Shiffman, 2015	USA	Original	Retrospective	120	60 (29–79)	63%	51%	81	39	1	SIM/SOF	NA	24 mo	[50]
Steinebrunner, 2018	Germany	Original	Retrospective	199	59 ± 10, (27–83)	67%	56%	152	47	1–4	SOF/LDV ± RBV, PTV/r/OMV/DSV ± RBV	NA	12 wks	[51]
Tahata, 2021	Japan	Original	Prospective	190	68 (40–87)	52%	57%	108	82	1–4	LDV/SOF, EBR/GZR, GLE/PIB, SOF/RBV, SOF/VEL ± RBV	No	12 wks	[52]
Terrault, 2016	USA	Original	Prospective	2255	60 (18–87)	60%	50%	917 (CHC924)	414	1	SOF/LDV/RBV	No	12	[53]
Tmu, 2019	India	Original	Retrospective	103	50 (29–82)	64%	18%	28 (CHC28)	47	1,2	SOF/RBV	No	12 wks	[54]
Verna, 2020	Worldwide	Original	Prospective	642	60 (25–89)	68%	65%	178	393	4	SOF/LDV, SOF/DCV, SOF/Vel, EBR/GZR	No	12 wks	[55]
Wei, 2020	China	Original	Prospective	222	58.9 ± 10.7	50%	9%	31 (CHC165)	26	1–6	OBV/PTV/r/DSV, SOF/DCV ± RBV, SOF/VEL ± RBV, SOF/RBV, EBR/GZR, DCV/ASV, GLE/PIB ± RBV	No	Median, 36 wks	[56]
Welzel, 2016	Germany	Original	Retrospective	485	57 (27–87)	66%	70%	223 (CHC97)	165	1–5	DCV/SOF ± RBV	No	12 wks	[57]
Belli, 2016	Europe	Original	Retrospective	103	54 (34–71)	68%	NA	0	103	1–3	SOF/RBV, SOF/DCV ± RBV, SOF/LDV ± RBV, SOF/SIM ± RBV	No	51.9 (32.9–67.4) wks	[58]
Belli, 2018	Europe	Original	Retrospective	36,382	54 (34–71)	69%	NA	0	36,382	NA	Protease inhibitor	NA	NA	[59]
Bittermann, 2021	USA	Original	Retrospective	8394	57 (53–61)	70%	NA	0	8394	NA	DAA	NA		[60]
Charlton, 2015	USA	Original	Prospective	108	59 (55–62)	67%	65%	0	108	1	SOF/LDV/RBV	No	NA	[9]
Curry, 2015	USA	Original	Prospective	267	58 (40–73)	70%	55%	0	267	1–6	SOF/Vel ± RBV	No	12 wks	[5]
Curry, 2016	Worldwide	Abstract	Prospective	667	NA	NA	NA	0	667	1/4	SOF/LDV	No	12 wks	[61]
Debnath, 2019	India	Abstract	Retrospective	62	Median 48	54%	NA	0	62	NA	SOF/DCV, LDV Vel ± RBV	No	24 wks	[62]
El-Sherif, 2018	Worldwide	Original	Retrospective	622	59 (54–62)	72%	NA	0	622	1–4	SOF/LDV/RBV, Vel/SOF ± RBV, SOF + RBV	No	255 (251–236) days	[63]
Flemming, 2017	USA	Original	Retrospective	47,591	56 (IQR, 51–61)	71%	NA	0	33,947	NA	DAA, PI	NA	NA	[64]
Foster, 2016	UK	Original	Prospective	409	54 (28–80)	73%	61%	0	409	1/3	SOF/LDV ± RBV, SOF/DCV ± RBV	No	12 wks	[26]
Gentile, 2019	Italy	Original	Prospective	89	72 (67–76)	46%	42%	0	89	1–4	SOF/LDV SOF/RBV SOF/DCV	No	11 months	[65]
Hanafy, 2019	Egypt	Original	Retrospective	160	51.4 ± 6.3	78%	0	0	160	4	SOF/DCV/RBV	No	29.3 ± 1.9 mo	[66]
Lin, 2018	China	Abstract	Retrospective	56	63.6 ± 8.1	39%	24%	0	56	1–3	DAA	NA	12.5 ± 7.3 months	[63]
Lourenco, 2021	Brazil	Original	Retrospective	85	56.13 ± 11.14	51%	54%	0	85	1–3	SOF/DCV ± RBV, SOF/SIM ± RBV	No	12–24 wks	[67]
Manolakopoulos, 2017	Greece	Abstract	Retrospective	126	59 ± 12.82	62%	56%	0	126	1–3	SOF, SOF/SIM, SOF/DSV, SOF/LDV, 3D/2D, SOF/VEL, EBR/GZP	No	12 wks	[68]
McCaughan, 2017	Australia	Original	Prospective	108	56 (51–61)	73%	40%	0	108	1–4	SOF/DCV ± RBV	No	12 wks	[69]
Modi, 2016	USA	Original	Prospective	42	58 (32–69)	74%	52%	0	42	1	SOF/SIM ± RBV	Yes	12 wks	[70]
Mohamed Essa, 2019	Egypt	Original	Retrospective	75	>60.20 (26.7)	69%	NA	0	75	NA	SOF/DCV, SOF/LDV ± RBV	No	6 mo	[71]
Sandmann, 2018	Worldwide	Original	Retrospective	35	55.5 ± 8.97	80%	54%	0	35	1–4	SOF/DCV ± RBV, SOF/SIM, SOF/LDV ± RBV, PTV/r/OBV/DSV	No	18 (IQR 8–29) mo	[72]

Abbreviations: CC, compensated cirrhosis; DC, decompensated cirrhosis; GT, genotype; DAA, direct antiviral agent; FU, follow-up; NA, not applicable; PI, protease inhibitor; SVR, sustained virologic response; EOT, end of treatment; IQR, interquartile range; Wks, weeks; mo, months; NA, not available; SOF, sofosbuvir; RBV, ribavirin; r, ritonavir; SIM, simeprevir; DCV, daclatasvir; LDV, ledipasvir; GLE, glecaprevir; PIB, pibrentasvir; EBR, elbasvir; GZR, grazoprevir; 2D, OBV/PTV/r; 3D, OBV/PTV/r + DSV; OBV, ombitasvir; PTV, paritaprevir; DSV, Dasabuvir; VEL, Velpatasvir. * References are found in the Appendix A.

**Table 2 jpm-12-01517-t002:** Summary of the estimated effect of the sustained virologic response of direct-acting antiviral (DAAs) in decompensated cirrhosis patients with chronic hepatitis C virus (HCV) infection.

Outcomes	No. of Studies	Pooled Event Rate	95% CI	I^2^ (%)
**SVR, overall**	49	0.86	0.83–0.88	68
**Age (years)**				
≥60	14	0.87	0.81–0.92	78.3
<60	35	0.84	0.82–0.87	66.1
**Previous interferon treatment**				
Yes	3	0.85	0.83–0.88	71.9
No	11	0.88	0.79–0.93	82.6
**Protease inhibitor based**				
Yes	6	0.85	0.75–0.91	78.7
No	37	0.86	0.84–0.89	74.4
**Genotype**				
GT 1	7	0.86	0.77–0.92	75.7
GT 3	3	0.8	0.45–0.95	50.5
GT 4	3	0.88	0.82–0.92	0
GT Mixed	35	0.85	0.82–0.88	77.1
**Study design**				
Prospective	25	0.86	0.83–0.89	70.9
Retrospective	23	0.84	0.80–0.88	73.1
**Industrial sponsorship**				
Yes	27	0.86	0.83–0.89	77.7
No	22	0.85	0.8–0.88	71
**Study region**				
Asia	9	0.87	0.77–0.93	57.6
USA	11	0.85	0.79–0.89	67.4
Europe	14	0.86	0.81–0.9	79.1
Others	15	0.86	0.8–0.88	71

**Table 3 jpm-12-01517-t003:** Summary of outcomes for patients with decompensated cirrhosis infected with hepatitis C virus (HCV) infection.

Outcomes	No. of Studies	Pooled Event Rate	95% CI	I^2^ (%)
SVR, overall	49	0.86	0.83–0.88	68
Adverse effect	13	0.55	0.31–0.77	96
Serious adverse effect	12	0.22	0.13–0.36	93
DAA discontinuation	11	0.06	0.04–0.08	52
Hepatocellular carcinoma	14	0.07	0.05–0.09	72.2
Mortality	28	0.06	0.04–0.08	77.6
Improvement in liver function	34	0.51	0.44–0.58	93
Worsening of liver function	20	0.16	0.12–0.21	73
Delisting from liver transplantation	6	0.16	0.05–0.40	98.8

**Table 4 jpm-12-01517-t004:** Summary of the estimated event rate in decompensated cirrhosis patients with chronic hepatitis C virus (HCV) infection depending on the use of protease-inhibitor-containing regimen.

Outcomes	No. of Studies	Pooled Event Rate	95% CI	I^2^ (%)
**Adverse effect**				
Yes	4	0.49	0.11–0.88	88.4
No	9	0.58	0.24–0.86	97.1
**Serious adverse effect**				
Yes	3	0.23	0.07–0.83	79.4
No	9	0.18	0.09–0.33	77.9
**Worsening of liver function**				
Yes	4	0.22	0.20–0.25	0
No	14	0.14	0.09–0.20	80.6
**Improvement in liver function**				
Yes	2	0.49	0–1.00	92.7
No	23	0.51	0.42–0.59	93.2
**Hepatocellular carcinoma**				
Yes	1	0.05	0.03–0.1	NA
No	11	0.07	0.04–0.1	77.8
**Mortality**				
Yes	5	0.05	0.04–0.07	0
No	21	0.06	0.04–0.09	82.4

Abbreviations: NA, not applicable.

## Data Availability

Clinical trial number: PROSPERO (International Prospective Register of Systematic Reviews), CRD42021241245.

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
