# Peer review of "Direct-Acting Antivirals for HCV Treatment in Decompensated Liver Cirrhosis Patients: A Systematic Review and Meta-Analysis"

_jpm, 2022, doi:10.3390/jpm12091517_

Round 1
Reviewer 1 Report
The present manuscript clarified the clinical outcomes and safety of PI-based DAA regimens in patients with decompensated cirrhosis (DC) via a systematic review and meta-analysis. This review arouses interest for readers and provides an important clue to treat DC patients with PI-based or non-PI-based DAA regimens. There are some issues that should be addressed or modified.
Abstract section:
(1) What does the first sentence (line 18-19; “Direct-acting….cirrhosis (DC).”) mean?
(2) Line 34-35: the phrase “relative low” compared to what?
(3) Line 35: the phrase “relative safe” compared to what?
Introduction section:
(4) Line 68: the phrase “2.2-fold greater” compared to what?
Materials and Methods section:
(5) Line 89: Could you please specify the first DAAs?
Results section:
(6) Authors should specify literatures corresponding to the 60 studies (Table 1) in the References section or Supplementary data.
Discussion section:
(7) Line 393-394: I don’t understand the phrase “between those treated ….AEs and mortality”.
(8) Line 400-401: the sentence should be corrected.
Author Response
-
Abstract section:
(1) What does the first sentence (line 18-19; “Direct-acting….cirrhosis (DC).”) mean?
- Thank you for the comment. The sentence has been revised as below.
- DAA therapy is known to clear hepatitis C virus infection in patients with decompensated cirrhosis (DC).
(2) Line 34-35: the phrase “relative low” compared to what?
- Thank you for the comment. The sentence has been revised as below.
- The HCC occurrence and mortality rates were low in patients with DC following DAA treatment.
(3) Line 35: the phrase “relative safe” compared to what?
- Thank you for the comment. The sentence has been revised as below.
- PI-based treatment in DC patients was relatively safe when compared to non-PI-based treatment.
Introduction section:
(4) Line 68: the phrase “2.2-fold greater” compared to what?
- Thank you for the comment. The sentence has been revised as below.
- In a phase 2 study comprising 40 patients (19 CTP class A and 21 CTP class B patients) treated with simeprevir, sofosbuvir, and daclatasvir for 12 weeks, the mean pharmacokinetic exposure to simeprevir was 2.2-fold greater after 8 weeks of treatment in CTP B than A.
Materials and Methods section:
(5) Line 89: Could you please specify the first DAAs?
- Thank you for the comment. I agree that the DAAs should be specified but we have not mentioned in the manuscript since all DAAs were included. After the reviewer’s suggestion, we have specified the DAAs in the manuscript.
- The included DAAs were sofosbuvir, sofosbuvir and ribavirin, simeprevir, daclatasvir; ledipasvir, glecaprevir, pibrentasvir, elbasvir, grazoprevir, dasabuvir, velpatasvir and ombitasvir/paritaprevir/ritonavir.
Results section:
(6) Authors should specify literatures corresponding to the 60 studies (Table 1) in the References section or Supplementary data.
- Thank you for the important comments. We have added the references in the supplementary data.
Discussion section:
(7) Line 393-394: I don’t understand the phrase “between those treated ….AEs and mortality”.
- Thank you for the important comments. We agree that this sentence is confusing and therefore, the sentence “The improvement in liver function was comparable between those treated with PI and those who did not have a similar rate of AEs and mortality, demonstrating that PI-based regimens can be considered for DC patients” have been changed as below.
- The improvement in liver function was comparable between those treated with PI and those who did not with a similar rate of AEs and mortality between these two groups, demonstrating that PI-based regimens can be considered for DC patients.
(8) Line 400-401: the sentence should be corrected.
- Thank you for the important comments. The sentence “When the improvement in liver function was assessed in overall DC patients with DC, we observed an improvement in 51% of these patients” has been changed as shown below.
- When the improvement in liver function was assessed in overall DC patients, we observed an improvement in 51% of these patients.
Reviewer 2 Report
This is an excellent systematic review addressing the important question of DAA use in decompensated cirrhosis, and thereby adding significantly to our knowledge of the safety and potential benefits risks of DAA treatment in this setting, both with PI-based and non-PI based regimens. A few minor suggestions follow:
1. Abstract line 18 - this sentence is incorrect : DAA therapy does not cause HCV infection. I suspect this is an editing error that is easily fixed.
2. Abstract line 18- I suggest using DAA as the abbreviation for "directly acting antiviral *therapy*" as this is the use the authors have chosen through the study.
3 Abstract line 32 and 34 Suggest omitting "The" from the start of each sentence, and line 36 omit "the" prior to liver function.
4. Suggest discussing whether any analysis was attempted comparing different PI based regimens, and if not, why this was not done, given the known issues with some regimens. This could also form part of the discussion, ie. given this was a pooled analysis and some PIs are known to be more problematic, the others are likely to be even safer than measured by the pooled rates.
Thanks for the opportunity to read this interesting piece of work.
Well done.
Author Response
1. Abstract line 18 - this sentence is incorrect: DAA therapy does not cause HCV infection. I suspect this is an editing error that is easily fixed.
- Thank you for your critical comment. The sentence has been revised as below.
- Direct-acting antiviral (DAA) therapy is known to clear hepatitis C virus infection in patients with decompensated cirrhosis (DC).
2. Abstract line 18- I suggest using DAA as the abbreviation for "directly acting antiviral *therapy*" as this is the use the authors have chosen through the study.
- Thank you for your comment. We have changed to DAA as the reviewer suggests.
3 Abstract line 32 and 34 Suggest omitting "The" from the start of each sentence, and line 36 omit "the" prior to liver function.
- Thank you for your comment. “The” has been removed from the start of each sentence and prior to liver function.
4. Suggest discussing whether any analysis was attempted comparing different PI based regimens, and if not, why this was not done, given the known issues with some regimens. This could also form part of the discussion, ie. given this was a pooled analysis and some PIs are known to be more problematic, the others are likely to be even safer than measured by the pooled rates.
- Thank you for the critical comment. Analysis of the outcome according to different PI based regimens is clinically important as the results could differ based on the PIs used. However, since our study focused on DC patients there was not many studies including PI regimen and furthermore, most of the regimens analyzed were categorized as mix regiment. These limitation has been mentioned in the discussion as reviewer has suggested.
- Our study had several limitations. First, the number of studies on PI-based DAA in DC patients was small, with inconsistent proportions of PI-based DAAs due to the limited number of trials that included DC patients therefore analysis for every single PI regimen was not available.